# Coordination of the N-Terminal Heme in the Non-Classical Peroxidase from *Escherichia coli*

**DOI:** 10.3390/molecules28124598

**Published:** 2023-06-07

**Authors:** Ricardo N. S. Oliveira, Sara R. M. M. de Aguiar, Sofia R. Pauleta

**Affiliations:** 1Microbial Stress Lab, UCIBIO—Applied Molecular Biosciences Unit, Department of Chemistry, NOVA School of Science and Technology, Universidade NOVA de Lisboa, 2829-516 Caparica, Portugal; 2Associate Laboratory i4HB—Institute for Health and Bioeconomy, NOVA School of Science and Technology, Universidade NOVA de Lisboa, 2829-516 Caparica, Portugal

**Keywords:** bacterial peroxidase, non-classical peroxidase, heme-peroxidases, tri-hemic enzyme, *c*-type heme, spectroscopic characterization, thermostability, circular dichroism

## Abstract

The non-classical bacterial peroxidase from *Escherichia coli*, YhjA, is proposed to deal with peroxidative stress in the periplasm when the bacterium is exposed to anoxic environments, defending it from hydrogen peroxide and allowing it to thrive under those conditions. This enzyme has a predicted transmembrane helix and is proposed to receive electrons from the quinol pool in an electron transfer pathway involving two hemes (NT and E) to accomplish the reduction of hydrogen peroxide in the periplasm at the third heme (P). Compared with classical bacterial peroxidases, these enzymes have an additional N-terminal domain binding the NT heme. In the absence of a structure of this protein, several residues (M82, M125 and H134) were mutated to identify the axial ligand of the NT heme. Spectroscopic data demonstrate differences only between the YhjA and YhjA M125A variant. In the YhjA M125A variant, the NT heme is high-spin with a lower reduction potential than in the wild-type. Thermostability was studied by circular dichroism, demonstrating that YhjA M125A is thermodynamically more unstable than YhjA, with a lower T_M_ (43 °C vs. 50 °C). These data also corroborate the structural model of this enzyme. The axial ligand of the NT heme was validated to be M125, and mutation of this residue was proven to affect the spectroscopic, kinetic, and thermodynamic properties of YhjA.

## 1. Introduction

After cyanobacterial photosynthesis shifted Earth’s atmosphere, bacteria adapted to grow in the presence of molecular oxygen [1]. During O_2_ reduction to H_2_O, intermediate species with the ability to hamper a vast number of biological processes are formed, which are named reactive oxygen species (ROS) [2]. ROS are highly reactive molecules, such as H_2_O_2_ and O_2_^•−^, capable of oxidizing amino acids, nucleic acids, lipids, and carbohydrates. Within the bacteria kingdom, ROS can be generated both endogenously and exogenously. H_2_O_2_ forms at approximately 15 μM/s in well-fed cells, whereas O_2_^•−^ has a slower formation rate of around 5 μM/s [3]. In *Escherichia coli*, cytoplasmic H_2_O_2_ is formed mainly outside the respiratory chain, but this pathway is responsible for the majority of the O_2_^•−^ released from the cytoplasm to the periplasm [3]. One of the primary scavengers of H_2_O_2_ in *E. coli* is the alkyl hydroperoxide reductase AhpCF, a highly efficient two-component NADH peroxidase that has a *k*_cat_/*K*_M_ of 4 × 10^7^ M^−1^·s^−1^ [4].

Some genes encoding for proteins related to the response to oxidative stress are regulated by OxyR, a regulator of hydrogen peroxide-inducible genes [5]. When the intracellular concentration of H_2_O_2_ rises above the OxyR threshold (around 0.1 μM), the transcription factor becomes activated [6,7]. In this scenario, the expression of the genes coding for catalase, KatG, and non-classical bacterial peroxidase YhjA are highly expressed.

YhjA is proposed to be a peroxidase face-sided to the periplasm with a transmembrane domain in the inner membrane that acts as a front-line defense against peroxides, especially those from exogenous sources. It has a double regulation by OxyR, as mentioned, and by an oxygen-responsive global transcription factor—fumarate and nitrate reductase (FNR) [8]. The characteristics of the *yhjA* promoter and the predicted function of YhjA suggest that transcription of the *yhjA* gene is likely to be restricted to specific environmental conditions in which oxygen is in low concentrations, such that some FNR activity is retained, and peroxides are present so that OxyR is active. This double regulation suggests that YhjA plays a specific role as a front-line defense against peroxidative stress in *E. coli*, possibly in micro-aerobic conditions or when transitioning from anoxic to oxic environments [9,10]. In this set of environmental conditions, as YhjA was shown to be an ineffective defense enzyme given the kinetic parameters that were determined [10], an additional role was proposed. Since the source of electrons resides in the quinol pool, it enables *E. coli* to respire at the cost of H_2_O_2_, thus using this molecule as an electron acceptor under anoxic conditions [11].

Genomic and transcriptomic analysis of *E. coli* planktonic cells and biofilms revealed differential *yhjA* expression patterns in different growth conditions. This gene has a higher expression level in biofilm compared to planktonic cells, and, in both cases, it has a higher expression level under anoxic conditions. Out of the genes encoding for the major *E. coli* H_2_O_2_ scavengers identified (*katG*, *katE*, *ahpC*, *tpx*, *btuE* and *bcp*), *yhjA* is the one with the highest fold change [12], agreeing with YhjA being mainly produced in anoxic environments [8] and being involved in H_2_O_2_ reduction [11].

Bacterial peroxidases are heme peroxidases that bind *c*-type hemes and are divided into two classes: classical, which harbor two *c*-type heme groups, and non-classical, with three *c*-type heme groups [13,14]. These enzymes are located in the periplasm of Gram-negative bacteria and receive electrons from small *c*-type mono-heme cytochromes and/or type-I copper proteins [15,16,17,18,19] in the case of the classical bacterial peroxidases or from the quinol pool in the case of the non-classical bacterial peroxidases [10,20].

The gene encoding for non-classical bacterial peroxidases has been mainly found in the genome of known pathogenic bacteria [21]. YhjA from *E. coli* [10,11], PerC from *Zymomonas mobilis* [22], and QPO from *Aggregatibacter actinomycetemcomitans* [20,23,24,25] are the only non-classical bacterial peroxidases isolated and characterized to date. Both YhjA and PerC have a putative transmembrane helix in the N-terminus, while QPO is confirmed to be a membrane-bound protein, with the three enzymes being classified as quinol-dependent bacterial peroxidases, as they use quinols as electron donors. PerC from *Z. mobilis* was shown to have an impact on the membrane NADH peroxidase activity during aerobic growth, and *perC* knockout *Z. mobilis* strain becomes sensitive to exogenous hydrogen peroxide [22].

The UV-visible spectrum of these enzymes in the oxidized state presents the typical cytochrome *c* absorption spectrum, with the Soret band around 410 nm and two additional absorption bands: (i) at 620 nm, a band characteristic of the presence of high-spin hemes and (ii) at 695 nm, a weak band attributed to a Met axial ligand of the N-terminal (NT) heme [10]. YhjA (purified without the transmembrane domain) has similar features to the classical bacterial peroxidases [13,14,26]: (i) a high-potential electron transferring heme (E heme), E′ of +210 mV, hexa-coordinated with a weakly bound methionine resulting in a thermal high-low spin equilibrium at room temperature; (ii) a low-potential peroxidatic heme (P heme), E′ of −170 mV; and (iii) requiring calcium ions to attain maximum activity. However, different structural and spectroscopic features are also present [10]: (i) the P heme is hexa-coordinated with a labile hydroxo ligand in an OH^−^/His coordination, and thus it is able to bind the substrate in the oxidized state; (ii) it has a third heme in the N-terminal domain—the NT heme—with an E′ of +133 mV, proposed to have a His/Met coordination; (iii) it does not require reductive activation to be fully active; and (iv) since it has a transmembrane domain, it is proposed to be able to receive electrons directly from the quinol pool.

In this work, different variants were generated to access the coordination of the NT heme. The spectroscopic properties, ability to bind the substrate and thermostability of the YhjA variants were accessed and compared with the wild-type YhjA.

## 2. Results and Discussion

### 2.1. Analysis of the Primary Sequence and Structural Model

The analysis of the primary sequence, using InterPro and PredictProtein tools, predicts the presence of a transmembrane domain at the N-terminus (comprising residues 7–30). The SignalP tool was used to predict the presence/absence of a signal peptide in YhjA’s primary sequence and to confirm that the transmembrane region identified previously would not be cleaved off. In the case of YhjA’s primary sequence, none of the major types of signal peptides were identified, classifying the sequence as having “other” signal peptide. Single and multiple transmembrane α-helix domains have been shown to act as uncleaved signal peptides that comprise a long hydrophobic chain [27]. These uncleaved signal peptides are characteristic of membrane proteins located in the bacterial inner membrane, as is the case for YhjA. Therefore, the first 30 residues of YhjA constitute a transmembrane region (*vide infra* in the analysis of the structural model).

The primary sequence alignment of YhjA with other primary sequences of classical and non-classical bacterial peroxidases was performed elsewhere [10]. Here, the alignment of YhjA with other non-classical bacterial peroxidases (Appendix A) will be analyzed. The sequences chosen were the three non-classical bacterial peroxidases that have been isolated and, to some extent, biochemically characterized [20,22,23,24,25] and the primary sequence of two other proteins identified in a phylogenetic tree as belonging to this class of enzymes [13]. In this alignment, some conserved residues are highlighted to discuss the structural model of YhjA, which was obtained using the AlphaFold2 CoLab (Figure 1). Although one should regard this structure as a mere model, its level of confidence is relatively high, as most of the backbone has a per-residue confidence score above 90, with three small regions having lower confidence: (i) the N-terminal helix and the N-terminal (NT) globular domain have some regions with a score between 70 and 90, and (ii) the loop linking the N-terminal helix and the other globular domains has a score between 50 and 70 (Appendix A). Thus, this structural model will be used to compare the spatial location of conserved residues and also to confirm that non-classical bacterial peroxidases have a globular hydrophilic region divided into two domains: (i) a C-terminal domain structurally homologous to classical bacterial peroxidases [13,28], confirming the hypothesis proposed based on the sequence alignment [10], and (ii) an N-terminal domain (designated NT domain) that is mainly composed of α-helices and binds an additional *c*-type heme. Overall, the hydrophilic region (solvent exposed) is mainly composed of α-helices and turns, with a single β-sheet in the C-terminal domain, similar to what is observed in classical bacterial peroxidases [28].

The sequence alignment shows the presence of three conserved -CX_2_CH- motifs highlighted in green, comprising the two cysteines that form the thioether bonds with the vinyl groups of the porphyrin ring. This sequence motif also comprises the proximal axial ligand of the three hemes (H63, H355, and H211 for the NT, E, and P hemes, respectively) (highlighted in green in Appendix A).

Three putative distal axial ligands of the NT heme were identified, M84, M125, and H134 (highlighted in grey in Appendix A), as they are conserved and well positioned in terms of distance in the primary sequence to the NT *c*-type heme motif. These three residues will be mutated to alanine to assess their effect on the spectroscopic properties of YhjA. However, upon analysis of the structural model obtained with AlphafFold2 CoLab (Appendix A), M125 is the only one in a position to be able to coordinate the NT heme iron, as its sulfur atom is vertically aligned with the imidazole group of H63, contrary to the other two residues M82 and H134 (Figure 1C and Appendix A). This observation validates the previous hypothesis that NT heme is Met/His hexa-coordinated by H63 and M125, according to spectroscopic data [10]. The structural model shows that the P heme lacks a residue coordinating the distal axial position (Figure 1C), which strengthens the spectroscopic data that pointed to OH^−^/His coordination; thus, the P heme is available for substrate binding in the oxidized state, as proposed in the literature [10]. The hydroxo group is a labile axial ligand proposed to fast exchange with H_2_O_2_ in its presence.

The transmembrane region (marked with a grey rectangle in Appendix A) does not show a high level of conservation among the aligned primary sequences. Nevertheless, this region was modeled as a helix, and it has a high hydrophobic character (colored in purple in Figure 1A), contrasting with the other domains (colored in green in Figure 1A), which is consistent with this helix being a transmembrane helix while the other domains are hydrophilic and will be present in the periplasm.

In this helix, there are three conserved residues—two tyrosine residues (Y19 and Y25) and one aspartate (D30) (numbering of the residues according to *E. coli* YhjA primary sequence)—that are highlighted in red in the sequence of the transmembrane domain (Figure 1 and Appendix A). Given their conservation and localization, a prime role was proposed to be their involvement in quinol-protein interaction (represented in Figure 1A). In fact, tyrosine, aspartate, tryptophane and histidine residues have been shown in other *E. coli* membrane proteins, such as fumarate reductase [29], cytochrome *bo*_3_ [30] and succinate dehydrogenase [31], to mediate the interaction between quinones and transmembrane domains and to stabilize the semiquinone.

The electron transfer between E and P hemes is proposed to be mediated by W250 [10,32] (highlighted in blue in Appendix A) in a charge-hopping mechanism, as proposed for other bacterial peroxidases [33]. Since non-classical bacterial peroxidases have an additional heme at the N-terminal domain, the mechanism through which the electrons flow from the NT heme to the E heme needs to be unraveled. In analogy to what is observed between E and P hemes, tryptophan residues might be involved in this electron transfer pathway. In fact, W118, conserved in some non-classical bacterial peroxidases (highlighted in blue in Appendix A), is well positioned to be able to mediate the electron transfer from the NT heme to the E heme (Figure 1B,C). Likewise, W135—conserved among all aligned primary sequences—could mediate the electron flow from the quinols to the NT heme (highlighted in blue in Appendix A and Figure 1B). Three additional aromatic residues—F53, Y72, and Y90 (highlighted in blue in Appendix A and Figure 1B)—are conserved among some aligned primary sequences and might also be involved in the electron transfer pathway from the quinols to the NT heme, given their location. 

Relative to the active site, the P heme pocket, there are two conserved residues, E270 and Q260 (Figure 1C and highlighted in purple in Appendix A), well positioned to be involved in the heterolytic cleavage of hydrogen peroxide at the O–O bond and in mediating the proton transfer required to complete the reduction of hydrogen peroxide to water, respectively. These residues have been identified in the classical bacterial peroxidases by site-directed mutagenesis to play such roles [32,34,35,36]. In common with the classical bacterial peroxidases, there are three conserved residues—T410, P412, and N235 (Figure 1C and highlighted in yellow in Appendix A)—well positioned to coordinate the calcium ion [35,36,37] through the oxygen atoms of their carbonyls or sidechain, respectively (in addition to four other water molecules that are not present in this structure, as it is a model). This also strengthens the requirement of calcium ions for maximum activity, modulating the p*K*_a_ of the E heme propionate groups [14].

Overall, the structural model supports the hypothesis of YhjA having a transmembrane domain comprising one α-helix, which is a highly hydrophobic region, and a globular region located in the periplasm, which presents several conserved features common to the classical bacterial peroxidases (Figure 1A). Nevertheless, the YhjA X-ray structure, as well as site-directed mutagenesis are required to confirm these hypotheses, especially the relative orientation of the domains, which will have an impact on the distances between residues and the electron transfer pathway. We will pursue these studies in the future.

### 2.2. Isolation and Characterization of YhjA Variants

Four variants of YhjA were constructed—M82A, M125A, and H134A to determine which of these residues would be coordinating the NT heme, and C59AC62A to produce YhjA without the NT heme. These variants were used in spectroscopic and kinetic assays to assess the role of this heme in protein stability and activity.

#### 2.2.1. Protein Isolation

YhjA wild-type (WT) and the four variants were initially produced in *E. coli* BL21(DE3) for spectroscopic characterization. In the case of the double mutant YhjA C59AC62A, the protein was produced and present in the periplasmic extract (Appendix A) but could not be purified, as it did not bind to the Strep-Tacting Sepharose matrix. One possible explanation is that in the absence of the NT heme, this domain adopts a different conformation, and the tag is no longer available for binding, or the domain is unstructured and prone to proteases, though protease inhibitors have always been added to the cell suspensions after being harvested.

The three variants—M82A, M125A, and H134A—were produced and purified in a single-step purification process with yields of 0.17, 0.11, and 0.21 mg per L of growth medium, respectively. The purity ratios (A_407nm_/A_280nm_) for YhjA WT, M82A, M125A, and H134A of the purified proteins were 3.2, 2.8, 3.0, and 3.0, respectively, which were considered pure, as no additional bands were present in the SDS-PAGE besides one (Appendix A). Protein and heme quantifications indicate that the variants were produced with three hemes per protein.

As the yield was very low and additional characterization was intended, production of WT and the M125A variant was optimized, and they were produced in *E. coli* C41 (DE3), which gave a higher yield—0.7 and 0.6 mg of pure protein per L of growth, respectively. These two proteins were produced with purity ratios of 3.3 and 2.9 for the WT YhjA and M125A variant, respectively. SDS-PAGE of the purified YhjA variants (Appendix A) shows no bands corresponding to other proteins, except for one band below YhjA that is attributed to a small level of proteolysis.

#### 2.2.2. UV-Visible Spectroscopy

The visible spectra of the YhjA variants were acquired in the oxidized and mixed-valence states (Figure 2). The spectra and the spectral changes observed between the two oxidation states of YhjA M82A and YhjA H134A are very similar (Figure 2A,C), while differences are clearly observed in the spectra of the YhjA M125A variant (Figure 2B).

In the case of YhjA M82A and YhjA H134A, the as-isolated spectra shows (i) the Soret band with a maximum at 407 nm, (ii) the Q bands are somewhat degenerated with a maximum at 532 nm, (iii) a band around 620 nm, attributed to the presence of a high-spin heme, and (iv) a weak band around 695 nm, attributed to the methionine coordinating the NT heme iron [10] (Figure 2A,C, solid line). Although the E heme also has a distal methionine ligand, it is weakly bound to the heme iron at room temperature (in a high/low-spin equilibrium) and thus does not contribute to the absorption band at 695 nm [10]. In the mixed-valence spectrum (Figure 2A,C, dashed line), the Soret shifts to 419 nm, due to the reduction of both E and NT hemes. A shoulder is visible at 402 nm, as the P heme remains oxidized and 6-coordinated with a hydroxo ligand. Therefore, the band at 620 nm is present, as the P heme remains high-spin, as observed for the YhjA WT [10]. The α and β bands become resolved at 553 nm and 524 nm, respectively. The band at 695 nm is no longer visible, as the NT heme is reduced.

The as-isolated spectrum of the YhjA M125A variant has the Soret and Q-absorption bands centered at slightly different wavelengths (405 nm and 531 nm, respectively) (Figure 2B, solid line), but the 570–720 nm region is different from that of YhjA WT [10] and the other variants. In common, there is an absorption band with a maximum at 620 nm, but it differs because there is an additional band centered at 655 nm (also present in the mixed-valence state), and the absorption band with a maximum at 695 nm is absent. The latter is expected, as the axial methionine ligand of the NT heme was mutated to an alanine, while the band at 655 nm is proposed to also arise from this heme, hinting that the heme iron is H_2_O/His hexacoordinated and not OH^−^/his coordinated [10,14,38,39].

In the mixed-valence spectrum of YhjA M125A (Figure 2B, dashed line), the Soret region has a main absorption at 403 nm with a shoulder at 418 nm, which is different from what was observed for the other variants (Figure 2A,C, dashed line). This indicates that there is another heme besides the P heme that remains oxidized (or partially oxidized), and the hypothesis is that it is the NT heme, which has no distal axial ligand; thus, its reduction potential is expected to be lower, as observed for hemes coordinated by solvent-derived molecules [40]. The fact that the absorption band at 655 nm is still present in the mixed-valence spectrum of YhjA M125A supports this hypothesis. Therefore, since sodium ascorbate no longer completely reduces the NT heme, its reduction potential must be lower than the value estimated in the WT, +133 mV [10], and lower than the reduction potential of sodium ascorbate (+66 mV, pH 7.0 [41]). In the Q-band region, the α- and β-bands become resolved with maxima at 553 nm and 524 nm, respectively. 

To sum up, the mixed-valence spectrum of YhjA M125A shows that the NT heme remains oxidized (Soret region and presence of 655 nm absorption band), and that the NT and P hemes are high-spin (NT heme with an H_2_O/His axial coordination and P heme with a OH^−^/His axial coordination).

Based on the heme content, the extinction coefficient of the Soret band in the oxidized state was determined to be 336 ± 6, 310 ± 10, and 330 ± 4 mM^−1^cm^−1^ for the YhjA M82A (at 407 nm), YhjA M125A (at 405 nm), and YhjA H134A (at 407 nm) variants, respectively.

#### 2.2.3. Spectral Changes in the Presence of Peroxides

The visible spectra of as-isolated YhjA WT (shown for comparison), YhjA M82A, YhjA M125A, and YhjA H134A variants were acquired in the presence of hydrogen peroxide. As shown previously, the P heme is high-spin and ready to bind the substrate in this oxidation state, so changes in the spectra—especially in the Soret band and in the 620 nm band—will indicate whether the oxoferryl species is being formed [10,33].

The spectra were acquired in the presence of 10-molar fold excess of hydrogen peroxide (Figure 3), and spectral changes were monitored for 10 min, to the point at which no further changes were observed. In the case of YhjA WT, M82A, and H134A, the addition of hydrogen peroxide induced a shift of the Soret band to 410 nm, decreasing its intensity, and disappearance of the band at 620 nm. This means that H_2_O_2_ is able to bind to the P heme and that oxoferryl species is being formed, as observed previously for YhjA [9,10,42] and the H71G variant of the classical bacterial peroxidase from *P. aeruginosa* [33].

In the case of the YhjA M125A variant, hydrogen peroxide induces a red shift in the Soret band (409 nm) and causes the disappearance of the band at 620 nm, similar to the other variants. However, the high-spin absorption band attributed to the NT heme, centered at 655 nm, is still present. This points to H_2_O_2_ not being able to bind to this heme iron, even if it is no longer coordinated by a methionine. A plausible hypothesis is that the heme cavity does not have enough space for H_2_O_2_ to bind, even if the methionine was replaced by an alanine residue, which has a shorter sidechain.

In the literature, it was reported that the *yhjA* knockout *E. coli* strain was more sensitive to both H_2_O_2_ and cumene hydroperoxide [8]. Thus, YhjA WT and YhjA M125A were also incubated with cumene hydroperoxide, but no changes in their visible spectra—neither in the Soret band nor in the band around 620 nm—were observed (Appendix A). This is an indication that cumene hydroperoxide does not bind to the high-spin P heme, or if it binds, it is only to a very low extent. This agrees with YhjA not being able to catalyze its reduction [9].

In conclusion, there is formation of an intermediate species of the catalytic cycle (Compound **I**) when hydrogen peroxide is present, which supports its use as the substrate of YhjA, though there are reports that peroxynitrite can also be a substrate [42]. However, those kinetic assays have not been performed under physiological conditions, nor was a *K*_M_ reported to be compared with the one determined for H_2_O_2_ [10].

#### 2.2.4. Enzymatic Activity of the YhjA WT and Variants 

Kinetic assays using ABTS^2−^ and hydroquinone as electron donors were performed to assess the effect of these mutations on the catalytic activity (Table 1). Clearly, the YhjA M125A variant has a lower specific activity, suggesting that an intact NT heme is required for maximum activity.

Considering both the spatial organization of the three hemes and their reduction potential, the order of electron transfer from the quinol pool is from quinol to NT heme, then to E heme and then to P heme (the catalytic center). Given that in the YhjA M125A variant, the reduction potential of the NT heme is lower than +66 mV, the electron transfer rate from the electron donor (ABTS^2−^ or hydroquinone) to the NT heme and from it to the E heme will be hampered, and consequently, the activity of the enzyme decreases.

A possible explanation for the fact that YhjA M125A retains some catalytic activity is that the artificial electron donors can donate electrons directly to the E or P heme, surpassing the electron transfer pathway involving the NT heme.

Moreover, it is known that heme proteins having a penta-coordinated heme can react non-specifically with hydrogen peroxide [43,44]. Thus, in the case of YhjA M125A, catalysis occurring at the NT heme, could, in principle, contribute to the observed re-oxidation rate of the electron donor in the presence of hydrogen peroxide. However, in this variant, the high-spin NT heme does not seem to be able to bind hydrogen peroxide, as the absorption band at 655 nm does not decrease in intensity in its presence (Figure 3C). Therefore, the ability of the NT heme to catalyze the hydrogen peroxide reduction should be significantly reduced.

In conclusion, the NT heme of YhjA is involved in the electron transfer pathway from the electron donors to the P heme.

### 2.3. Circular Dichroism of YhjA WT and Variants

Circular dichroism (CD) spectra of YhjA WT were acquired in the far-UV region (between 190 nm and 260 nm) to determine the secondary structure content of YhjA WT and compare it with that of the structural model. Given the range of wavelengths of the collected data, the analysis can only differentiate the percentages of α-helix, β-sheet, and other structural motifs [45], with no distinction between the types of helixes and β-sheets. The CD spectra of YhjA variants (M82A, M125A, and H134A) was also acquired in this region to assess the effect of the mutation on the folding status and on the thermostability. 

#### 2.3.1. Folding Status and Secondary Structure Content

The CD spectra of YhjA WT and M82A, M125A, and H134A variants in the far-UV region are presented in Figure 4. The CD spectra of the studied YhjA variants almost superimpose with YhjA WT. The positive peak at 195 nm, present in all the CD spectra, indicates that the proteins are folded; thus, the difference in the specific activity observed previously for the YhjA M125A variant is not a reflection of the enzyme being unfolded. Moreover, as the CD spectra of variants superimpose with that of YhjA WT, none of the point mutations change the secondary structure content of the protein. 

The two negative peaks at 208 nm and 222 nm are indicative of a higher content of α-helix, in agreement with the structural model obtained (Figure 1B). The θ_208nm_/θ_222nm_ ratio for all the CD spectra is 1.2, which is higher than 1, indicating that YhjA and the variants are not completely α-helix but have a high content of this secondary structure, similar to what was observed for the classical bacterial peroxidase from *Pseudomonas aeruginosa* [46].

The secondary structure content of YhjA WT was estimated using BeStSel [47,48] and DichroWeb [49,50] servers (Table 2), and 2Struc [51] was used to determine the percentage of the secondary structural motifs from the structural model generated with AlphaFold2 CoLab (considering YhjA without the transmembrane domain). The values obtained are similar, and the experimental data support the structural model that predicts YhjA to be mainly helical (Figure 2B). Given that the level of confidence is lower in the NT domain relative to the C-terminal domain (Appendix A), the difference between the content of α-helix and β-sheet estimated from the CD data and the model might reflect differences in the structure of the NT domain.

#### 2.3.2. Thermostability

The thermostability of YhjA WT and the M125A variant was studied using CD. Both proteins were subjected to a temperature ramp from 10 to 94 °C (Figure 5A,B), and the CD spectra acquired at each temperature were analyzed to examine the change in the secondary structure during the unfolding process (Appendix A). The profile of the secondary structure content of YhjA WT and the M125A variant with increasing temperature differs from one another. In the case of YhjA WT, there is a defined temperature range when the structure changes significantly (between 40 and 50 °C), whereas for the YhjA M125A variant, it seems to be a more gradual process that starts at lower temperatures. These profiles also show that for both proteins, the α-helix content decreases with concomitant unfolding while the β-sheet content increases (Appendix A). This different behavior is clearly induced by the mutation of residue 125 from a methionine to an alanine, as this is the only difference between the two proteins.

To better understand the thermodynamic differences in the unfolding process between YhjA WT and the M125A variant, the mean residue ellipticity at 222 nm as a function of temperature was fitted using the equations for a two-state transition of a monomer from a folded to an unfolded state (Figure 5C,D). However, the spectra obtained after the temperature ramp (in blue) do not superimpose with the initial spectra for both YhjA WT and the YhjA M125A variant, indicating that the unfolding process is only partially reversible. The ellipticity at 195 nm decreased from 11.7 to 6.5 mdeg, and 14.3 to 8.0 mdeg, for YhjA and the M125A variant, respectively, while the θ_208nm_/θ_222nm_ ratio is 1.3 for both proteins, but higher than before.

YhjA WT and the M125A variant have estimated melting temperatures of 50 °C and 43 °C, and enthalpies of unfolding of 239 kJ·mol^−1^ and 136 kJ·mol^−1^, respectively. These data indicate a less stable structure for the YhjA M125A variant, with a decrease of 6 °C in the melting temperature and a lower enthalpy of unfolding, meaning that less energy is required to disrupt the secondary structure of the protein. Furthermore, the melting temperature of 50 °C obtained by CD is similar to the value obtained previously by differential scanning calorimetry (DSC) [17] (provides information on the overall unfolding process), and it is close to the one estimated for the *P. pantotrophus* bacterial peroxidase (T_M_ 52 °C, ΔH 175 kJ·mol^−1^) [52]. The value of enthalpy of unfolding estimated by CD (239 kJ·mol^−1^) is lower than the one estimated by DSC (~600 kJ·mol^−1^). The difference in enthalpy could be related to the absence of calcium ions in the buffer used in the CD analysis since it is known that calcium stabilizes the structure of several bacterial peroxidases [14,26,52] and because the process is not completely reversible. Therefore, true thermodynamic parameters cannot be determined from the CD data [53].

The estimated melting temperatures for the YhjA M82A and H134A variants were 52 and 51 °C, respectively (Appendix A). Thus, as these values are similar to the one estimated for YhjA WT (50 °C), the point mutations do not affect the thermostability of these variants.

## 3. Materials and Methods

### 3.1. Bioinformatic Analysis

The primary sequence of YhjA was analyzed with two online software, InterPro 90.0 (https://www.ebi.ac.uk/interpro/, accessed on 18 March 2023) and PredictProtein (https://predictprotein.org/, accessed on 18 March 2023) [54], to do a first assessment of the protein domains and features and later aligned with primary sequences of other non-classical bacterial peroxidases from different bacteria using Clustal-Omega (EMBL-EBI, https://www.ebi.ac.uk/Tools/msa/clustalo/, accessed on 18 March 2023) [55]. SignalP 6.0 (https://services.healthtech.dtu.dk/services/SignalP-6.0/, accessed on 18 March 2023) was used to predict the presence/absence of a signal peptide sequence [56]. AlphaFold2 CoLab (https://alphafold.ebi.ac.uk/, accessed on 18 March 2023) was used to obtain a structural model of YhjA [57,58]. E and P hemes were positioned according to the structure superimposition with the coordinates PDB ID 2C1V [28], the NT heme was oriented so that it was coordinated by its axial ligands, and the cysteines were at a distance to form a thioether bond with the vinyls of the heme (Appendix A). The heme coordinates chosen for the P heme have a saddle structure, while for the other two, the heme has a ruffled distortion, usually observed in hexa-coordinated hemes involved in electron transfer [59,60]. All structural images were prepared using Discovery Studio Visualizer. 2Struc (https://2struccompare.cryst.bbk.ac.uk/index.php, accessed on 18 March 2023) was used to estimate the percentages of different secondary structure motifs of the structural model [51].

### 3.2. Generation of the Variants

The three variants were generated (M82A, M125A, H134A and C59AC62A) by site-directed mutagenesis using the NZY mutagenesis kit (NZYTech, Lisbon, Portugal) and the pET22b_*yhjA* construct, previously produced [17], as a template. The plasmid was amplified with NZYproof enzyme (NZYTech, Lisbon, Portugal) with the appropriately designed primers (Appendix A) and using 65 °C for annealing and 7 min for extension, followed by *Dpn*I hydrolysis of methylated DNA prior to transformation. After plasmid propagation in *E. coli* XL-1 blue (Merck, Lowe, NJ, USA) and isolation using a mini-prep kit (NZYtech, Lisbon, Portugal), the sequence was confirmed by sequencing (Stabvida, Caparica, Portugal). The produced proteins will have an N-terminal StrepII-tag with a TEV protease cleavage site for tag removal after purification.

### 3.3. Protein Production and Isolation

The soluble YhjA WT, as well as the variants, were obtained as follows. *E. coli* C41 (DE3) competent cells (Merck, Lowe, NJ, USA) were co-transformed with pEC86 and either pET22b_yhjA or pET22b_yhjAM125A, while *E. coli* BL21 (DE3) competent cells were co-transformed with pEC86 and either pET22b_yhjAC59AC62A, pET22b_yhjAM82A or pET22b_yhjAH134A. pEC86 harbors the *ccm* operon to produce all the machinery for *c*-type heme biosynthesis and maturation [61] and confers chloramphenicol resistance. Four to five colonies of transformants were used to inoculate 100 mL of LB medium supplemented with 100 µg/mL of ampicillin and 30 µg/mL of chloramphenicol and incubated at 37 °C and 210 rpm overnight. Fresh medium (500 mL of LB supplemented with the same antibiotics) was inoculated with 2% of the pre-inoculum and grown at 37 °C and 210 rpm (Shel Lab, Cornelius, OR, USA) until an OD of 1.5 at 600 nm was reached. At this point, the cells were harvested at 3500× *g*, 6 °C for 20 min (Beckman Coulter-Avanti J-26 XPI, Brea, CA, USA) and resuspended in half the volume of fresh 2xYT medium supplemented with the same antibiotics and 185 µM of FeCl_3_ (Merck, Lowe, NJ, USA). The cells were stabilized for 1 h at 37 °C and 120 rpm, after which isopropyl β-D-1-thiogalactopyranoside (OmniPur, Merck, Lowe, NJ, USA) was added to a final concentration of 0.5 mM to induce YhjA production, and the cell suspension was grown at 30 °C and 120 rpm for 20 h. The cells were harvested at 7500× *g*, 6 °C for 10 min (Beckman Coulter-Avanti J-26 XPI, Brea, CA, USA) and resuspended in 50 mM Tris-HCl, pH 7.6 with protease inhibitor (Roche cOmplete™, EDTA-free Protease Inhibitor Cocktail, Basel, Switzerland). 

The periplasmic fraction was obtained with 5 freeze-thaw cycles. To eliminate spheroplasts and cell debris, the suspension was centrifuged at 40,000× *g*, 6 °C for 45 min (Beckman Coulter-Avanti J-26 XPI, Brea, CA, USA), adding DNase I (Roche, Basel, Switzerland) prior to centrifugation. The purification of soluble YhjA or YhjA variants was achieved with affinity chromatography, using a 5 mL StrepTrap column (Cytiva, Marlborough, MA, USA) equilibrated with 100 mM Tris HCl, pH 7.6, 500 mM NaCl. After applying the periplasmic fraction, the non-adsorbed proteins were eluted with 5 column volumes of the same buffer. YhjA or YhjA variants were eluted with 2.5 mM D-desthiobiotin (Sigma-Aldrich, Merck, Lowe, NJ, USA) in 100 mM Tris-HCl, pH 7.6, 500 mM NaCl. The fractions containing YhjA were concentrated over a 30 kDa Amicon (Merck, Lowe, NJ, USA), and the buffer was exchanged in the same Amicon to 20 mM HEPES, pH 7.5 (Sigma-Aldrich, Merck, Lowe, NJ, USA).

Protein and heme quantification were performed using the modified Lowry method [62] and the pyridine hemochromagen method [63], respectively. For protein quantification, horse heart cytochrome *c* was used as the standard protein.

### 3.4. Spectroscopic Characterization

UV-visible spectra were acquired on a Shimadzu UV-1800 spectrophotometer at room temperature and aerobic conditions using quartz cells with a 10 mm optical path between 350 nm and 750 nm. Spectra were acquired for both YhjA WT and variants in the as-isolated and mixed valence states. YhjA in the mixed-valence state was obtained by reduction with a solution of sodium ascorbate (Asc) (Alfa Aesar, Thermo Fisher Scientific, Havergill, MA, USA) and 2,3,5,6-tetramethyl-1,4-phenylenediamine 3,6-diamino-durol (DAD) (Fluka Chemie GmbH, Buchs, Switzerland) prepared in 20% ethanol to a final concentration of 1 mM and 5 μM, respectively. For this, the enzyme with the Asc/DAD solution was incubated for one hour at room temperature. 

Spectra of as-isolated YhjA WT and variants in the presence of 10-molar fold hydrogen peroxide (Sigma-Aldrich, Merck, Lowe, NJ, USA) or cumene hydroperoxide (Sigma-Aldrich, Merck, Lowe, NJ, USA) were also acquired after incubation with each compound until no spectral change was observed (10 min).

Circular dichroism (CD) spectra in the far-UV region were acquired in an Applied Photophysics Chirascan™ qCD spectrometer for YhjA WT and its M82A, M125A, and H134A variants using a 0.2 mm path length cuvette with a total volume of 170 μL. The protein samples were prepared in 20 mM sodium or potassium phosphate, pH 7.0, by buffer exchange in a 30 kDa Amicon (Merck, Lowe, NJ, USA), with successive centrifugations at 3900× *g*, 4 °C. The concentration of the samples of YhjA WT and its M82A, M125A, and H134A variants was 8.3 μM, 12.5 μM, 10.9 μM, and 11.8 μM, respectively. These concentrations were determined using the absorbance at 205 nm, and the extinction coefficient was calculated using the primary sequence composition [64] (ε_205nm_ per residue: 3953 M^−1^cm^−1^, 3948 M^−1^cm^−1^, 3948 M^−1^cm^−1^, and 3941 M^−1^cm^−1^, for YhjA WT and M82A, M125A, and H134A variants, respectively). The CD data were reported in mean residue ellipticity ([θ]_MRE_) as a function of wavelength. The [θ]_MRE_ is the CD raw data (in deg) corrected for the concentration of the protein solution using Equation (1),
(1)$${\left[\theta \right]}_{MRE}=\frac{\theta \left(deg\right)\times MRW}{10\times L\times C}$$
in which MRW (mean residue weight) is the molecular mass of YhjA WT, or of the variants (M82A, M125A or H134A) divided by the number of residues, the concentration of the protein in the sample is (C), in gmL^−1^, and the pathlength of the cell is (L), in cm.

Spectra acquired at 25 °C from 190 nm to 260 nm are the average of three spectral acquisitions, with a bandwidth and a step-size of 1 nm, and acquired with a time per point of 3 s. The far-UV data analysis to determine the secondary structure content was performed with the servers BeStSel (https://bestsel.elte.hu/, accessed on 18 March 2023) [47,48] and Dichroweb (http://dichroweb.cryst.bbk.ac.uk/, accessed on 18 March 2023) [49,50].

Far-UV CD spectra were acquired at different temperatures between 10 °C and 94 °C. The spectra were acquired with 1s per point in intervals of 2 or 5 °C after incubating the sample for one minute at each temperature. The characterization of YhjA’s unfolding process was analyzed using the Gibbs-Helmholtz method—to fit the change of CD at a single wavelength as a function of temperature, considering a two-state transition of a monomer from a folded to unfolded state, and considering that the heat capacity of the folded and the unfolded states are equal, ΔCp = 0 [53]. For the Gibbs-Helmholtz method, the experimental data were fitted using Equations (2)–(5) and using the SOLVER add-in program of Microsoft Excel (version 16.7). The molar ellipticity at any given temperature (T), [θ]_T_, is given by Equation (2)
(2)$${\left[ϴ\right]}_{T}=\alpha \times \left({\left[ϴ\right]}_{F}-{\left[ϴ\right]}_{U}\right)+{\left[ϴ\right]}_{U},$$
in which the fraction folded at any given temperature, α, is given by Equation (3)
(3)$$\alpha =\frac{K}{1+K}$$
and the folding constant, K, at any given temperature is given by Equation (4)
(4)$$K=\exp\left(\frac{-\Delta G}{R\times T}\right)$$
and ΔG is given by Equation (5)
(5)$$\Delta G=\Delta H\times \left(1-\frac{T}{T_{M}}\right)$$
in which T_M_ is the temperature at which the fraction folded, α, is 0.5.

### 3.5. Kinetic Assays

The enzymatic activity of the YhjA WT and its variants was assessed with two different electron donors: (i) 2,2′-azino-bis-(3-ethyl-benzthiazoline-6-sulphonic acid (ABTS^2−^) oxidation was monitored by the increase in absorbance at 420 nm (ε_420nm_ = 36 mM^−1^·cm^−1^) over time [9,10]; (ii) benzene-1,4-diol (hydroquinone) oxidation forming benzoquinone was monitored by the increase in absorbance at 260 nm (ε_260nm_ = 5.4 mM^−1^·cm^−1^) [9,10]. The assays were performed at 25 °C in 10 mM HEPES pH 7.5, 10 mM NaCl, and 1 mM CaCl_2_, containing 3 mM ABTS^2−^ or 100 μM quinol, and 1 mM H_2_O_2_, and initiated with the addition of YhjA variants (15–45 nM). To ensure the presence of only hydroquinone (the reduced form), 70 mg of zinc powder was added to 20 mM quinol solution (in 1.7 mL ethanol with 0.18 M HCl), and this solution was degassed and maintained on ice during the assays (method described by [65]). All the solutions used in the kinetic assays with hydroquinone as electron donor were degassed, and a constant argon flow was used during each assay to maintain the anoxic conditions. The observed initial rates, *v*_0_, were determined in the first seconds of the re-oxidation curve by subtracting the electron donor auto-oxidation rates to determine the real initial rates.

## 4. Conclusions

Non-classical bacterial peroxidases have been characterized to some extent, but little is known about the role of the additional third heme (NT heme), at the N-terminal domain, and how it intervenes in the electron transfer pathway and catalytic efficiency, as well as the function of the transmembrane helix. Spectroscopic analysis of soluble YhjA without the transmembrane domain and its variants on putative axial ligands of the NT heme (M82, M125, H134) show key differences. Nevertheless, it was not possible to isolate the enzyme without the NT heme (YhjA C59AC62A variant). 

Regarding visible spectroscopy, YhjA M125A variant’s NT heme is in a high-spin state and must have a reduction potential lower than 60 mV, as it cannot be reduced by sodium ascorbate. These data, in combination with the analysis of the structural model discussed here in detail, show that the NT heme is Met/His coordinated by M125.

UV-visible spectra acquired in the presence of hydrogen peroxide and cumene hydroperoxide show that only H_2_O_2_ can form the oxoferryl species, inducing changes in the Soret band and the disappearance of the 620 nm band and thus binding to the P heme. In addition, since the absorption band observed in the spectrum of the YhjA M125A at 655 nm is not affected by the presence of hydrogen peroxide, it cannot bind to the NT heme. Thus, the NT heme in the YhjA M125A variant is not able to catalyze the reduction of hydrogen peroxide at a significant rate, even if it is known that such activity has been observed for other high-spin hemes in non-true peroxidases [43,44]. The reduction of the activity observed for the YhjA M125A variant strengthens the hypothesis that the NT heme is involved in the electron transfer pathway from the electron donor to the P heme. 

Moreover, the YhjA M125A variant is less thermodynamically stable, having a T_M_ of 43 °C and a ΔH of 136 kJ/mol, when compared to YhjA WT, which has a T_M_ of 50 °C and a ΔH of 239 kJ/mol.

The mutation of the axial ligand of the NT heme affects the spectroscopic features of YhjA, validating this heme in the protein’s function, as it also affects its activity. Further kinetic studies using YhjA WT and the M125A variant will validate the involvement of the NT heme in the electron transfer pathway and in the catalytic cycle, contributing to the elucidation of the NT heme’s role in non-classical bacterial peroxidases and its requirement to receive electrons from the quinol pool.

## Figures and Tables

**Figure 1 molecules-28-04598-f001:**
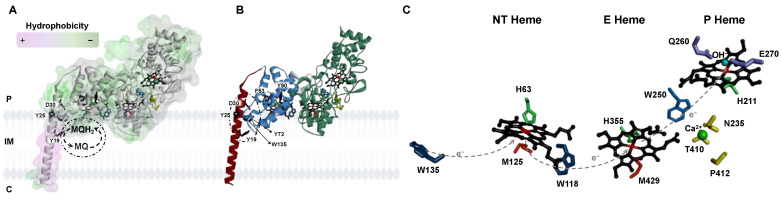
Structural model of the non-classical bacterial peroxidase from *E. coli*, YhjA, highlighting its cellular localization and key residues in electron transfer and quinol binding. (**A**) YhjA structure highlighting the degree of hydrophobicity of the surface, colored from low (green) to high (purple) level of hydrophobicity. (**B**) Polypeptide chain colored by domain (N-terminal domain, blue; C-terminal domain, green; transmembrane helix, red). (**C**) Coordination of NT, E, and P hemes in YhjA structural model. Legend: P-periplasm, IM-Inner membrane, C-cytoplasm, MQ-menaquinone, MQH_2_-menaquinol. Hemes are colored in black; residues proposed to stabilize the interaction between the protein and the semiquinone are colored in black; residues proposed to mediate electron transfer to and between hemes are colored in blue; histidine and methionine residues coordinating the heme irons are colored green and orange, respectively; in yellow, calcium-binding residues; calcium ion is a green sphere, and OH^−^-coordinating P heme is a blue sphere. W135, W118, and W250 are blue; Q260 and E270 are purple. Dashed arrows indicate the proposed electron flow mediated by the tryptophan residues. The structure of YhjA was obtained using AlphaFold2 CoLab. Heme groups were added manually as described in Materials and Methods. Structure was prepared in Discovery Studio Visualizer.

**Figure 2 molecules-28-04598-f002:**
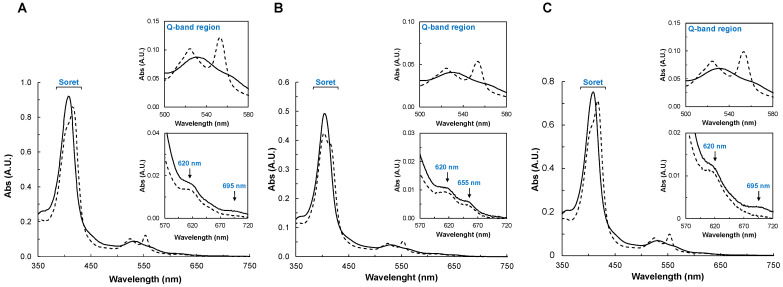
Visible spectra of YhjA M82A (**A**), M125A (**B**), and H134A (**C**) variants in 20 mM HEPES, pH 7.5, in the as -isolated state (black line) and mixed-valence state (dashed black line). In the insets are shown the Q-band region (upper panel) and the 620 nm band region (lower panel). Spectra were recorded between 350 nm and 750 nm for 2.7 µM YhjA M82A, 1.6 µM YhjA M125A, and 2.3 µM YhjA H134A.

**Figure 3 molecules-28-04598-f003:**
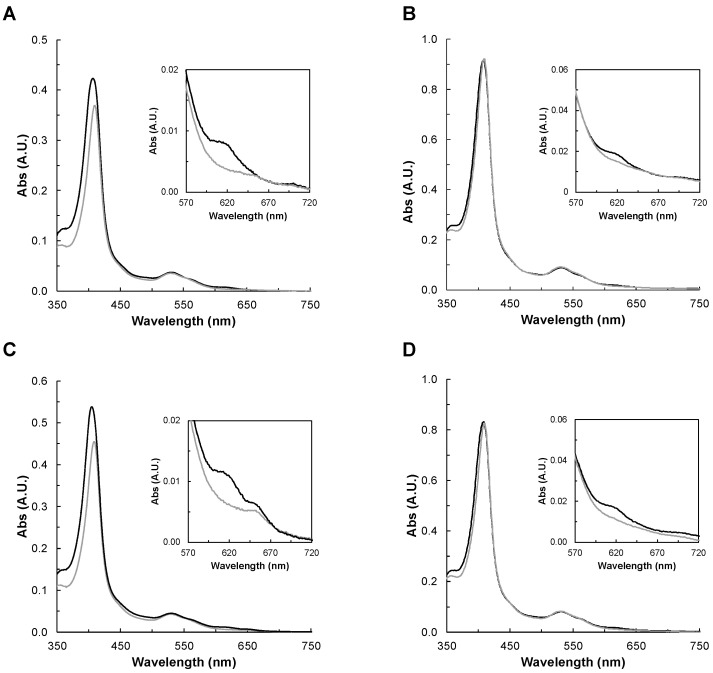
Visible spectra of YhjA WT (**A**) and M82A (**B**), M125A (**C**), and H134A (**D**) variants in the as-isolated state in 20 mM HEPES, pH 7.5 in the presence of 10-fold hydrogen peroxide after 10 min incubation. Enzyme before and after addition of hydrogen peroxide is represented by black or grey lines, respectively. In the insets are shown the 620 nm band region. Spectra were recorded between 350 nm and 750 nm for 1.7 µM YhjA WT, 2.7 µM YhjA M82A, 1.8 µM YhjA M125A, and 2.6 µM YhjA H134A.

**Figure 4 molecules-28-04598-f004:**
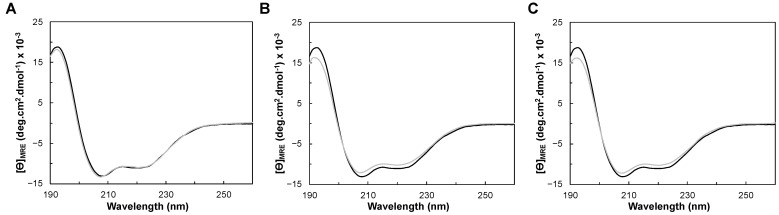
Far-UV circular dichroism spectra of (**A**) YhjA WT (black) and YhjA M125A variant (grey), (**B**) YhjA WT (black) and YhjA M82A variant (grey) and (**C**) YhjA WT (black) and YhjA H134A variant (grey), in 20 mM phosphate buffer pH 7.0 at 25 °C.

**Figure 5 molecules-28-04598-f005:**
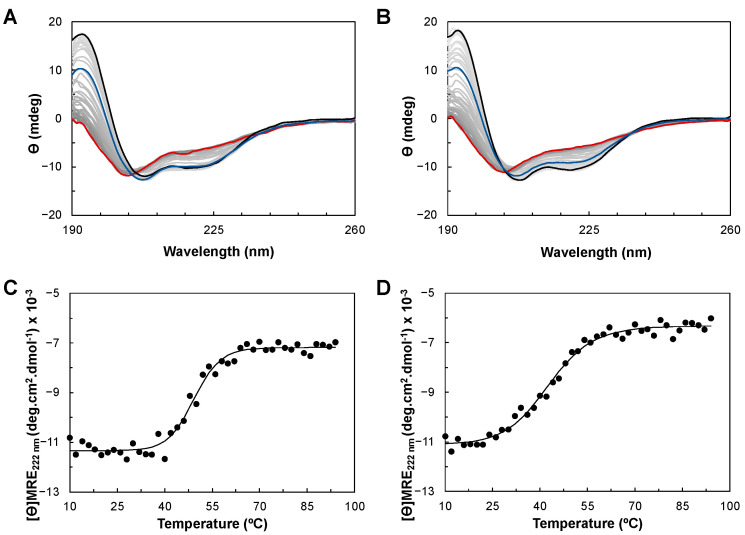
Thermal denaturation of YhjA WT (Panels **A** and **C**) and YhjA M125A variant (Panels **B** and **D**), monitored by circular dichroism in the far-UV region. Panels (**A**) and (**B**): Spectra acquired during a temperature ramp between 10 °C (black) and 94 °C (red); spectra acquired at 20 °C after the temperature ramp in blue. Panels (**C**) and (**D**): Mean residue ellipticity ([ϴ]_MRE_) at 222 nm as a function of temperature for (**C**) YhjA WT and (**D**) YhjA M125A variant. The data were fitted to a two-state transition according to Equations (1)–(4).

**Table 1 molecules-28-04598-t001:** Specific activity of YhjA (nmol/s/mg) and its variants using either ABTS^2−^ or hydroquinone as electron donor.

	YhjA	YhjA M82A	YhjA M125A	YhjA H134A
ABTS^2−^	110 ± 3	111 ± 12	68 ± 14	110 ± 18
Hydroquinone	54.6 ± 0.1	54 ± 13	32 ± 11	56 ± 13

**Table 2 molecules-28-04598-t002:** Secondary structure content of YhjA WT and variants estimated from the CD spectra and structural model. For DichroWeb and 2Struc, the values are the averages, or the range of values obtained from the different methods, respectively.

	Data from BeStSel/DichroWeb or 2Struc		
	α-Helix (%)	β-Sheet	Other
YhjA WT	31/36	14/14	55/50
YhjA structural model	40–50	1–14	46–50

## Data Availability

Additional data are presented in Appendix A, and any other data are available upon request.

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
