# Peer review of "Coordination of the N-Terminal Heme in the Non-Classical Peroxidase from Escherichia coli"

_molecules, 2023, doi:10.3390/molecules28124598_

Round 1

Reviewer 1 Report

The authors present a study of the coordination of the NT heme in the non-classical peroxidase from E. coli. The study is based on spectroscopic data of specific variants. Besides the spin state of the heme irons, they also stud the thermal stability (via CD spectroscopy) and the peroxidase activity. The study is interesting and can be published, after taking into account the below comments.

My main comment about this work is that I miss a small discussion part in which the current findings are linked to what is known about peroxidases and the role of key residues in the heme pocket. I think the manuscript would benefit from such a part, since it now remains very much on a level of mere describing observations.

Further comments are:

P3, line 133-136: the authors mention that the 3 conserved residues in the transmembrane region are proposed to be involved in the quinol-protein interaction. On what basis is this assumption based? Please mention a reference where this is described, or explain.

I assume that the modelling was done not including the heme cofactors. I suggest that the authors discuss also potential limits of the post insertion of the hemes, especially in confirming the presence or lack of distal residues (e.g. the claim in lines 193-196 that the model confirms a lack of a distal amino-acid residue ligation to the heme iron.

Line 194-196: the P-heme coordinates hydroxide, so this means this ligand needs to be first replaced by the substrate. Hence, I am a bit surprised that the P-heme is described as being ready to bind substrates.

Figure 2: I think the readability of the paper would improve if the absorption peaks you refer to in the text would be clearly indicated in the figure.

The authors speculate that the redox potential of the NT heme in M125A will have changed. This is definitely probable. I would however suggest that the authors determine the redox potential.

Figure 3: I would use different colors to make it easier for the reader to know which spectrum is of which condition. The legend mentions 1, 3, 5, 10 min for (B) and (D), but this is very hard to see in the actual spectra.

Line 287, Figure 3: for M82A and H134A it is very hard to see from the figure that there is a shift of the Soret band. I suggest putting either an inset or a figure in the supplementary material with a blow-up of the top region of the Soret peaks.

Line 300-303: I am confused by this explanation. How can you know that the high-spin heme is binding water? Also, the sentence ‘However, the high-spin attributed to this heme is still present’ is not really a scientific way of putting this. I guess you mean that the absorption peaks that you attribute to this heme are still present.

Table 1: I find it interesting that the M125A mutant still shows significant activity. This actually indicates that the role of M125 is relatively moderate. Can this be linked to knowledge about other peroxidases? Or cytochromes?

Line 356: I agree that the model agrees with the observation of the CD spectra for the alpha-helices, but the difference between the beta-sheets percentage for the model is significantly lower than what is derived from the data. Please comment on the potential dangers that this entails for interpretations based on the model only (see my earlier comment).

Figure S4: how was the secondary structure content determined, with BestSel or DichroWeb ?(I assume the former).

Thermostability: why is only the thermostability determined for the M125A variant. Are the other ones behaving exactly the same as the wild-type? If yes, please mention this and include the data in the supplementary material. If not, what does this say about the protein?

Small comments/typo’s

Line 33: underlined in Figure S1 – since the line is on top of the sequence, I would put ‘marked with a black line’

P4 figure 1B: the labels of F53 and Y90 are very hard to see. Please indicate these residues better, e.g. with an arrow. The color of the key residues is also not clearly seen. Maybe chose a more vibrant color?

Line 365: defined temperature instead of define temperature

Line 369: supplementary instead of Supplemeentary

Figure S1: I think it would be more practical to also indicate the putative distal ligand residues in a specific color, instead of the letter N. Maybe put the letter in a color, if you do not want to mark it with a colored stripe.  

Line 511, equation (4): the symbols were replaced by ‘?’, but this is probably a problem of the pdf generation. Please check in final version.

Author Response

We would like to thank the reviewers for taking time to carefully read and comment on our manuscript. Their valuable comments and questions were all taken into consideration and the document was modified accordingly, which helped improve this final version.

Our answers are in blue and the modified text has been provided in each answer, where applicable. The modified text in the main manuscript is also highlighted in blue.

We believe that this modified manuscript is now at a stage that can be accepted for publication in Molecules.

Reviewer #1 comments and answers:

The authors present a study of the coordination of the NT heme in the non-classical peroxidase from E. coli. The study is based on spectroscopic data of specific variants. Besides the spin state of the heme irons, they also stud the thermal stability (via CD spectroscopy) and the peroxidase activity. The study is interesting and can be published, after taking into account the below comments.

My main comment about this work is that I miss a small discussion part in which the current findings are linked to what is known about peroxidases and the role of key residues in the heme pocket. I think the manuscript would benefit from such a part, since it now remains very much on a level of mere describing observations.

We would like to thank the reviewer for taking the time to thoroughly reading our manuscript. We have taken into consideration all the comments to modify the submitted manuscript, and our answers to the main concerns are below.

Further comments are:

  1. P3, line 133-136: the authors mention that the 3 conserved residues in the transmembrane region are proposed to be involved in the quinol-protein interaction. On what basis is this assumption based? Please mention a reference where this is described, or explain.

This statement is based on (i) conservation of these residues and (ii) special localization in the structural model, and (iii) such residue types have shown to be involved in quinone binding and stabilization of semiquinone in other membrane proteins. Some of these other studies are referenced in the manuscript. This part of the analysis of the primary sequence was rewritten for clarity.

  1. I assume that the modelling was done not including the heme cofactors. I suggest that the authors discuss also potential limits of the post insertion of the hemes, especially in confirming the presence or lack of distal residues (e.g. the claim in lines 193-196 that the model confirms a lack of a distal amino-acid residue ligation to the heme iron.

The AlphaFold2 CoLab was used to obtain a model of YhjA’s structure. At the time, the database for the automatically generated structures was not available. In addition, the program Alphafill was also not available. Therefore, the procedure that is now better described in Materials and Methods section, and discussed was to use the structure of the heme E and P of classical peroxidases in the active state (which backbone completely superimpose). The position of a c-type heme is not so difficult to be done manually, as its orientation must be correct to form the thioether bonds between the sulfur atom from the thiol of the cysteine sidechain and the vinyl group of the heme (the two cysteine residues are the ones in the motif CXXCH). In addition, the orientation of the heme must be such for the histidine in this motif to be coordinating the heme iron. The other axial ligand of the heme would then be possible to be identified, as will be located vertically on the other side of the heme iron.

The hemes that were added were chosen so that: (i) in the case of NT heme and E heme, which are Met/His coordinated it should be a raffled heme, while for (ii) P heme which is OH/His coordinated is a saddled heme (see references about this in the study by S.R. Pauleta et al. 2001 and 2007, DOI: 10.1021/bi002870z and 10.1021/bi702486d). The heme distortion cannot always be observed by X-ray unless a high-resolution structure is available. These data arise from some of the vibration modes of the heme, which are attained by the resonance Raman.

  1. Line 194-196: the P-heme coordinates hydroxide, so this means this ligand needs to be first replaced by the substrate. Hence, I am a bit surprised that the P-heme is described as being ready to bind substrates.

In bacterial peroxidases (see ref. DOI: 10.1016/j.ccr.2023.215114), P heme is proposed to be coordinated by a labile hydroxo ligand, as there is a band at 620 nm in its visible spectrum. The exchange of the hydroxo for the substrate is very fast, and it is why it is mentioned that the active site is ready-to-bind the substrate. However, we agree with the reviewer that wording “ready-to-bind” the substrate may not be the most accurate and leads to misinterpretation. The manuscript was modified to “P heme is available for substrate binding in the oxidized state,…” (to indicate that it can bind the substrate without the need to significantly change the P heme pocket).

  1. Figure 2: I think the readability of the paper would improve if the absorption peaks you refer to in the text would be clearly indicated in the figure.

This figure was modified to account for the reviewer suggestion.

  1. The authors speculate that the redox potential of the NT heme in M125A will have changed. This is definitely probable. I would however suggest that the authors determine the redox potential.

We agree with the reviewer that it would improve this manuscript, however at this time we cannot perform this experiment because it must be performed inside a glove box (due to the low reduction potential of the heme), which is under repair. We do not consider that knowing this value changes any of the conclusions, as it is clearly shown that the reduction potential of NT heme is lower than before (+ 133 mV versus SHE), as it cannot be reduced with sodium ascorbate. The decrease in the reduction potential is also consistent with the heme being in a high-spin state, as these hemes have a lower reduction potential.

  1. Figure 3: I would use different colors to make it easier for the reader to know which spectrum is of which condition. The legend mentions 1, 3, 5, 10 min for (B) and (D), but this is very hard to see in the actual spectra.

Line 287, Figure 3: for M82A and H134A it is very hard to see from the figure that there is a shift of the Soret band. I suggest putting either an inset or a figure in the supplementary material with a blow-up of the top region of the Soret peaks.

This figure was modified and now it only shows the spectra after 10 min incubation with hydrogen peroxide for simplicity. We believe that it is now easier for the reader to see the shift in the Soret band. The disappearance of the high-spin at 620 nm is clearly observed in all the spectra. Relative to the band at 655 nm, which we have attributed to the water-molecule coordinating the M125A NT heme, there is only a small effect which is an indication that hydrogen peroxide may not be binding to this heme because the size of heme pocket is smaller than the one of P heme. The change observed can be just because the band at 620 nm disappeared, as the spectrum was not deconvoluted for the different contributions.

  1. Line 300-303: I am confused by this explanation. How can you know that the high-spin heme is binding water? Also, the sentence ‘However, the high-spin attributed to this heme is still present’ is not really a scientific way of putting this. I guess you mean that the absorption peaks that you attribute to this heme are still present.

The high-spin band in the visible spectra can be either at 620 nm or 640 nm. In each case, it has been shown that it corresponds to a hidroxo or water molecule, respectively. These conclusions were taken from the analysis of the nIR MCD spectra, together with EPR and visible spectra of bacterial peroxidases and myoglobin (discussed in Adv Inorg Chem(2000) 51 163 and 10.1016/j.bbabio.2018.03.008).

The sentence was modified to clarify that we were referring to the absorption bands.

  1. Table 1: I find it interesting that the M125A mutant still shows significant activity. This actually indicates that the role of M125 is relatively moderate. Can this be linked to knowledge about other peroxidases? Or cytochromes?

The authors consider that the decrease in the reduction potential of the NT heme hampers it to receive electrons efficiently from the electron donors, and to mediate electron transfer to E heme. Thus, the activity observed can be explained considered that we are using soluble artificial electron donors that might be donating electrons directly to E or P heme, surpassing the NT heme. This is now better discussed in the manuscript.

High-spin hemes can react non-specifically with hydrogen peroxide. However, the authors believe that this is not occurring at the NT heme as it cannot bind this molecule (as explained before). This discussion was also added to the manuscript.

  1. Line 356: I agree that the model agrees with the observation of the CD spectra for the alpha-helices, but the difference between the beta-sheets percentage for the model is significantly lower than what is derived from the data. Please comment on the potential dangers that this entails for interpretations based on the model only (see my earlier comment).

The CD is used to corroborate the model structure. The authors acknowledge the dangers in overusing the model to take conclusions, especially for domains for which there are no significant homologous structures, which is the case of the NT domain (we are characterizing the soluble domain, so the transmembrane domain was not considered for the CD analysis). We have introduced the confidence of the model, which is presented in supplementary materials, and modified the manuscript to explain that the difference in percentage of secondary structure can be in this extra-domain which has low homology to other proteins in the database and a lower confidence score in the modeling. In addition, the percentage of secondary structure from a structural model is calculated using different algorithms, which give slightly different values (the range of the values is now given).

  1. Figure S4: how was the secondary structure content determined, with BestSel or DichroWeb ?(I assume the former).

The secondary structure content presented in Figure S4 was determined in BeStSel. This is now clearly mentioned in the legend of this figure.

  1. Thermostability: why is only the thermostability determined for the M125A variant. Are the other ones behaving exactly the same as the wild-type? If yes, please mention this and include the data in the supplementary material. If not, what does this say about the protein?

The thermostability of the M125A was studied because different spectroscopic and kinetic properties were observed for this variant. Therefore, to be sure that there was no stability issue with this protein its CD spectra and thermostability was studied. In the case of the other two mutants, no difference in the spectroscopic and kinetic properties was observed relative to the wild-type and therefore their thermostability was not studied at the time. We have acquired the CD spectra of these two other mutants (shown in Figure 4 and in Figure S6 in Supplementary Materials), showing that they are folded at room temperature. Moreover, these mutations do not affect their thermostability (Figure S6).

Small comments/typo’s

  1. Line 133: underlined in Figure S1 – since the line is on top of the sequence, I would put ‘marked with a black line’

This figure was modified. The transmembrane region is now included in a grey rectangle. The sentence in the manuscript was modified accordingly.

  1. P4 figure 1B: the labels of F53 and Y90 are very hard to see. Please indicate these residues better, e.g. with an arrow. The color of the key residues is also not clearly seen. Maybe chose a more vibrant color?

This figure was modified in order for the labels and color of the residues to be easily distinguishable.

  1. Line 365: defined temperature instead of define temperature
  2. Line 369: supplementary instead of Supplemeentary

These two typos were corrected.

  1. Figure S1: I think it would be more practical to also indicate the putative distal ligand residues in a specific color, instead of the letter N. Maybe put the letter in a color, if you do not want to mark it with a colored stripe.

The putative distal ligand residues of the NT heme are now indicated as a grey bar.

  1. Line 511, equation (4): the symbols were replaced by ‘?’, but this is probably a problem of the pdf generation. Please check in final version.

This was a problem with the generation of the pdf, as in the word file it is OK. We hope that it is corrected in this version.

Reviewer 2 Report

The paper addresses important basic questions, although it lacks biophysical techniques such as electron paramagnetic spectroscopy (EPR) or anaerobic titrations used in their important previous work (Cláudia S. Nóbrega, Bart Devreese, Sofia R. Pauleta, YhjA - An Escherichia coli trihemic enzyme with quinol peroxidase activity, Biochimica et Biophysica Acta (BBA) - Bioenergetics, Volume 1859, Issue 6, 2018, Pages 411-422, ISSN 0005-2728, https://doi.org/10.1016/j.bbabio.2018.03.008).

The conclusions made are limited by the techniques used to characterize the mutants. If those are not available, maybe the authors have a strain with deleted YhjA that can be complemented with the mutant plasmids and observe some change in the bacteria physiology.

Please use a consistent nomenclature (heme or haeme).

Please cite the original reference to plasmid pEC86 (Engin Arslan, Henk Schulz, Rachel Zufferey, Peter Künzler, Linda Thöny-Meyer, Overproduction of the the Bradyrhizobium japonicum c-Type Cytochrome Subunits of the cbb3 Oxidase in Escherichia coli, Biochemical and Biophysical Research Communications, Volume 251, Issue 3, 1998, Pages 744-747, ISSN 0006-291X,https://doi.org/10.1006/bbrc.1998.9549.)

Author Response

We would like to thank the reviewers for taking time to carefully read and comment on our manuscript. Their valuable comments and questions were all taken into consideration and the document was modified accordingly, which helped improve this final version.

Our answers are in blue and the modified text has been provided in each answer, where applicable. The modified text in the main manuscript is also highlighted in blue.

We believe that this modified manuscript is now at a stage that can be accepted for publication in Molecules.

Reviewer #2 comments and answers:

The paper addresses important basic questions, although it lacks biophysical techniques such as electron paramagnetic spectroscopy (EPR) or anaerobic titrations used in their important previous work (Cláudia S. Nóbrega, Bart Devreese, Sofia R. Pauleta, YhjA - An Escherichia coli trihemic enzyme with quinol peroxidase activity, Biochimica et Biophysica Acta (BBA) - Bioenergetics, Volume 1859, Issue 6, 2018, Pages 411-422, ISSN 0005-2728, https://doi.org/10.1016/j.bbabio.2018.03.008).

We would like to thank the reviewer for taking the time to thoroughly reading our manuscript. We have taken into consideration all the comments to modify the submitted manuscript, and our answers to the main concerns are below.

  1. The conclusions made are limited by the techniques used to characterize the mutants. If those are not available, maybe the authors have a strain with deleted YhjA that can be complemented with the mutant plasmids and observe some change in the bacteria physiology.

We thank the reviewer for this suggestion. However, we believe that this falls outside of the scope of the manuscript, but it is planned to be performed. We consider that physiologically is more relevant to show the quinol activity with the full-length enzyme, which is being carried out in the lab.

  1. Please use a consistent nomenclature (heme or haem).

This inconsistency of the use of nomenclature present in the Figures and text was corrected in the revised version of the manuscript.

  1. Please cite the original reference to plasmid pEC86 (Engin Arslan, Henk Schulz, Rachel Zufferey, Peter Künzler, Linda Thöny-Meyer, Overproduction of the the Bradyrhizobium japonicum c-Type Cytochrome Subunits of the cbb3 Oxidase in Escherichia coli, Biochemical and Biophysical Research Communications, Volume 251, Issue 3, 1998, Pages 744-747, ISSN 0006-291X,https://doi.org/10.1006/bbrc.1998.9549.)

This reference was added to the reference list, and in fact should have not been forgotten.

Round 2

Reviewer 2 Report

Great news! The paper has been enhanced and important factors have been thoroughly considered.